# FutureMorph: Toward Predicting Future Deformation Fields in Longitudinal Imaging

**Samah Khawaled**[1,3]                                                   SK3446@CORNELL.EDU
**Rudolf L. M. van Herten**[1,3]                                   RLV4001@MED.CORNELL.EDU
**Rachit Saluja**[1,2,3]                                                  RS2492@CORNELL.EDU
**Mert R. Sabuncu**[1,2,3]                                       MSABUNCU@CORNELL.EDU

[1] *Cornell Tech*
[2] *School of Electrical and Computer Engineering, Cornell University*
[3] *Department of Radiology,Weill Cornell Medicine*

**Editors:** Accepted for publication at MIDL 2026

## Abstract

Understanding how anatomy evolves over time is essential for tracking disease progression, quantifying risk, and studying healthy development and aging. Existing approaches either synthesize future images without modeling geometry or perform longitudinal registration that requires follow-up scans. We introduce **FutureMorph**, a framework that treats longitudinal forecasting as metadata-conditioned prediction of future diffeomorphic deformation fields. Given a baseline image (e.g., a brain MRI) and subject-level metadata (age, sex, and clinical variables), FutureMorph predicts time-indexed, subject-specific diffeomorphic deformation fields that explicitly capture future anatomical change. We employ a metadata-conditioned U-Net to estimate stationary velocity vector fields, which are integrated into smooth diffeomorphisms and applied using a spatial transformer to synthesize future images. Experiments on the OASIS-3 dataset show that our framework produces clinically meaningful predicted deformations and realistic future scans, capturing age- and interval-dependent trajectories. Our work provides a new perspective for longitudinal imaging studies by unifying image synthesis and deformation modeling.

**Keywords:** Longitudinal MRI, Deformation Field, Neurodegeneration, Aging, Conditional Modelling, Deep Learning

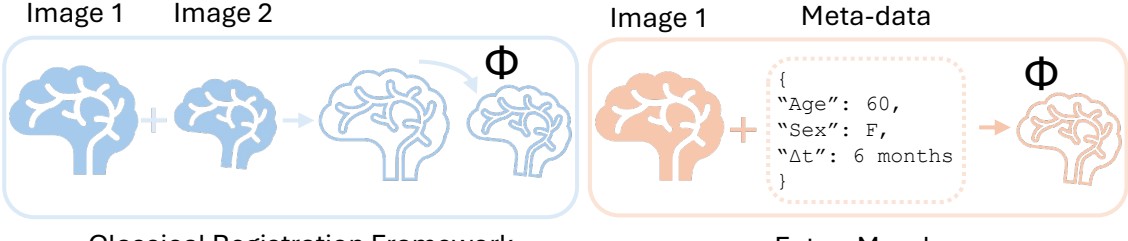

Figure 1: Comparison between classical image registration and FutureMorph. Traditional registration estimates a deformation $\Phi \in \mathbb{R}^3$ between two observed images $(I_{t_0}, I_{t_1})$. In contrast, FutureMorph predicts the future deformation $\Phi_{t_0 \to t_1}$ using only a baseline image and subject-level metadata (e.g., age, sex, and time interval), enabling deformation forecasting without requiring a follow-up scan.

## 1. Introduction

Quantifying structural changes in longitudinal imaging studies is essential for monitoring development, aging, disease progression, and treatment response. However, obtaining detailed descriptions of anatomical evolution typically requires access to follow-up scans, which are often unavailable or difficult to acquire in practice. Many existing approaches therefore focus on synthesizing future images, or on registration methods that estimate deformation fields between observed time points. While both strategies provide valuable information, they do not explicitly address the task of predicting geometrical change when only a baseline image and subject-level metadata are available. Recent advances in deep generative modeling, including Variational Autoencoders (VAEs) (Sauty and Durrleman, 2022), Generative Adversarial Networks (GANs) (Ravi et al., 2022; Pombo et al., 2023; Xia et al., 2021), diffusion models (Yoon et al., 2023), and latent diffusion frameworks (Puglisi et al., 2024), have been used to synthesize future scans or visualize disease trajectories. These methods produce realistic predictions but typically lack explicit representations of the geometric transformations underlying anatomical change. In many clinical or research settings, however, deformation fields provide more interpretable and anatomically grounded information than raw synthesized images. Motivated by this gap, we introduce a new formulation of longitudinal imaging prediction:

> **Motivation**
>
> Given an image at time $t_0$ and auxiliary metadata (e.g., age, sex, treatment, time interval), can we estimate a plausible future deformation field mapping $t_0$ to a later time $t_1$?

Our perspective provides a mechanism both for understanding future geometric changes by predicting future brain morphology. We generate follow-up images through spatial transformation of a baseline scan and additional patient specific meta data only. Classical longitudinal analysis relies heavily on image registration, where deformation fields quantify atrophy, growth, and other structural changes over time (Bartel et al., 2019; Shuaibu et al., 2025). Modern deep learning–based registration methods (Adrian V. Dalca et al., 2019; Balakrishnan et al., 2019; de Vos et al., 2019) can estimate diffeomorphic transformations efficiently, but they require both the baseline and the follow-up image. The presence of a fixed (future) image is precisely what makes the deformation estimation problem well-posed: it provides the target geometry that the deformation must align to. In contrast, FutureMorph addresses a setting where the fixed image does not exist (Figure 1). From this viewpoint, our framework can be interpreted as "registration with a missing fixed image": the model must infer the future deformation—mapping the baseline scan at time $t_0$ to an unseen scan at time $t_1$, using only the baseline anatomy and auxiliary subject-level metadata. This formulation showcases the practical role of FutureMorph: it acts as a predictive surrogate for the missing follow-up image, enabling downstream tasks that normally rely on registration, such as longitudinal change quantification, future-anatomy visualization, and disease-progression modeling.

In this work, we introduce FutureMorph, a framework that predicts future diffeomorphic deformation fields in longitudinal imaging using only a baseline scan, the time interval, and

subject-level metadata. FutureMorph predicts stationary velocity fields (SVFs), which are integrated to yield diffeomorphic transformations. Subject-specific metadata are encoded through a lightweight MLP and injected into the U-Net (Ronneberger et al., 2015) backbone using FiLM modulation (Perez et al., 2018), enabling the predicted deformation dynamics to adjust to demographic and clinical factors while preserving architectural flexibility. We adopt a VoxelMorph-style U-Net (Balakrishnan et al., 2019; Adrian V. Dalca et al., 2019), although our conditioning mechanism is architecture-agnostic.

Evaluations on OASIS-3 (LaMontagne et al., 2019) show that FutureMorph captures clinically meaningful aging-related trajectories. It also produces more heterogeneous and biologically realistic deformations than time-only SVF integration, which often yields overly smooth or implausible transformations. While the model is designed to forecast deformations, the resulting diffeomorphisms can be applied to the baseline image to visualize predicted future anatomy when needed. Our contributions are summarized as follows:

- We introduce FutureMorph, a formulation of longitudinal imaging focused on predicting future diffeomorphic deformation fields rather than future images alone.
- We design a metadata-conditioned deformation model that uses FiLM-modulated SVF prediction to incorporate age, time interval, and clinical factors into the deformation trajectory.
- We demonstrate that metadata conditioning yields more realistic, clinically aligned deformation dynamics than conventional time-only SVF extrapolation.

## 2. Related Works

Generation of future deformation fields is an important task in longitudinal neuroimaging, as deformations provide a direct and interpretable description of anatomical change over time. Recent models such as (Kim et al., 2022; Wang et al., 2025) use diffusion models to generate plausible deformations. However, they remain tied to the standard registration paradigm, in which both baseline and follow-up images must be provided. Generative approaches for longitudinal MRI—spanning diffusion (Yoon et al., 2023), latent diffusion (Puglisi et al., 2024), and structurally guided variants incorporating atrophy or brain-age priors (Litrico et al., 2024)—can synthesize realistic future scans. However, they primarily operate in image space and therefore do not yield explicit deformation fields that characterize the underlying geometric change. To address this limitation, Wu et al.(Wu et al., 2025) introduce a latent geodesic diffusion framework that models distributions of geodesic deformations conditioned on text prompts. Their method relies on a separate autoencoder-based registration module to infer latent geodesic trajectories, and is demonstrated only on 2D slices, which limits its applicability to full volumetric forecasting. In contrast, our approach directly predicts 3D diffeomorphic deformation fields using only a baseline image and subject metadata, without requiring paired scans or auxiliary registration networks. This formulation reframes longitudinal prediction around a central question: *Can we infer a realistic candidate deformation field from a single baseline scan and metadata alone?*

## 3. Methods

Let $I_{t_0}, I_{t_1} \in \mathbb{R}^{H \times W \times D}$ denote 3D images (e.g., MRI scans) acquired at two time points $t_0$ and $t_1$ ($t_1 > t_0$), with time interval $\Delta t = t_1 - t_0$. In our setting, we are given the

baseline image $I_{t_0}$ along with tabular metadata $d$, which can include variables such as $\{\Delta t, \text{age}, \text{sex}, \text{disease class}, \text{disease score such as mini-mental state exam, or MMSE}\}$. Our objective is to predict a plausible $\Phi$ that maps $I_{t_0}$ to $I_{t_1}$.

In our work, we parameterize a diffeomorphic $\Phi^t : \mathbb{R}^3 \to \mathbb{R}^3$ via the ordinary differential equation (ODE):

$$\frac{d}{dt}\Phi^t = V(\Phi^t), \quad \Phi^0 = I_d, \tag{1}$$

where $I_d$ is the identity transformation, and $V$ is a stationary velocity field (SVF). The final deformation is then obtained as $\phi^1$ by integrating $V$ over $t \in [0, 1]$. Estimation of $\phi$ is formulated as an energy minimization problem:

$$\hat{\Phi} = \arg\min_\Phi \ \mathcal{S}\big(I_{t_1}, \Phi \circ I_{t_0}\big) + \lambda \, \mathcal{R}(\Phi), \tag{2}$$

where $\phi \circ I_{t_0}$ denotes warping the moving image $I_{t_0}$ with $\Phi$, and $\mathcal{S}(\cdot, \cdot)$ measures similarity to the fixed image $I_{t_1}$. The regularization term $\mathcal{R}(\Phi)$ enforces smoothness, ensuring that $\Phi$ remains diffeomorphic. The trade-off between similarity and regularity is controlled by $\lambda > 0$. We predict the velocity field $V$ using a U-Net, $f_\theta$, with trainable weights $\theta$, taking the baseline image and metadata as input: $V = f_\theta(I_{t_0}, d)$.

Our model is based on the VoxelMorph backbone (Balakrishnan et al., 2019; Adrian V. Dalca et al., 2019)[1]. The predicted $V$ is then integrated via scaling-and-squaring (SS) (Ashburner, 2007), to approximate (1) and obtain $\phi$. During training (when $I_{t_1}$ is available), $\theta$ is optimized using the loss in (2). Figure 2 illustrates the overall scheme of our approach. The SVF-estimation module extracts multi-scale features from the baseline MRI, while the metadata encoder $\mathcal{E}$ embeds subject-specific information such as age, sex, and clinical variables. These embeddings are injected into the model via conditioning to guide the SVF prediction. This enables the network to learn deformation patterns conditioned on baseline MRI and metadata, allowing prediction of plausible longitudinal deformations even in the absence of the follow-up scan $I_{t_1}$. To further illustrate the advantages of our learnable conditioning mechanism, we examine two ways of parameterizing and using the SVF within FutureMorph, each corresponding to a different assumption about how anatomical change evolves over time. The first approach conditions the SVF estimation on the full metadata. The second, referred to as FutureMorph-S, integrates a stationary SVF up to the time interval $\Delta t$, while the SVF network receives only the remaining metadata (age, sex, etc.; dashed arrow in Figure 2).

### 3.1. FutureMorph: Time-conditioned SVF Prediction

To incorporate subject-specific information, we condition the U-Net on metadata $d = \{\Delta t, \text{age}, \text{sex}, \text{diagnosis}, \text{MMSE}\}$. The metadata encoder maps $d$ into a low-dimensional embedding vector $\mathbf{z} \in \mathbb{R}^h$ by combining learned embeddings for categorical variables (sex, diagnosis) with MLP layers for continuous variables (age, $\Delta t$, MMSE). Formally, $\mathbf{z} = \mathcal{E}(d)$, where $E(\cdot)$ denotes the metadata encoder. We apply Feature-wise Linear Modulation (FiLM) (Perez et al., 2018) to inject the metadata embedding $\mathbf{z}$ into the SVF estimation network. Conditioning is applied to all layers of both encoder and decoder. Let $\mathbf{x}^{(l)} \in \mathbb{R}^{B \times C_l \times D \times H \times W}$

---

1. Architecture details are provided in Appendix A

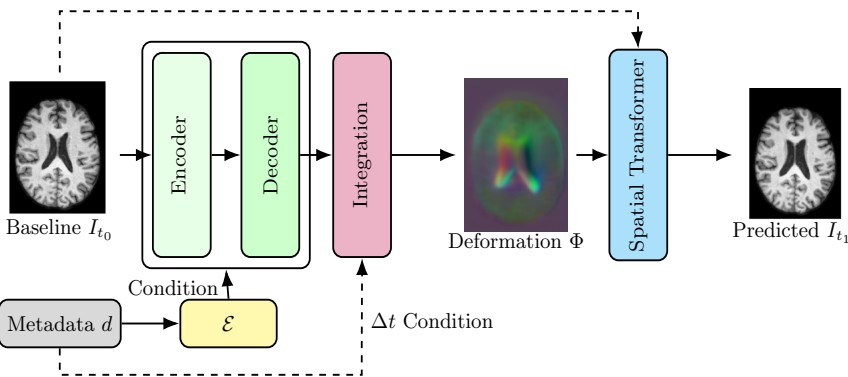

Figure 2: **FutureMorph**: Overview of our framework. Given a baseline MRI $I_{t_0}$ and meta-data vector $d$, the SVF is predicted by a U-Net backbone, which is conditioned on embeddings from the metadata encoder $\mathcal{E}$. The predicted SVF is then integrated to yield a future diffeomorphic deformation, $\Phi$, which is applied to the baseline image to generate $I_{t_1}$.

denote the activation of the $l$-th decoder layer, where $B$ is the batch size, $C_l$ is the number of feature channels at layer $l$, and $(D, H, W)$ are the spatial dimensions.

For each layer $l$, FiLM generates a scale vector $\gamma^{(l)}(\mathbf{z}) \in \mathbb{R}^{C_l}$ and a bias vector $\beta^{(l)}(\mathbf{z}) \in \mathbb{R}^{C_l}$ from the metadata embedding through learned linear projections:

$$\gamma^{(l)}(\mathbf{z}) = W_\gamma^{(l)}\mathbf{z} + b_\gamma^{(l)}, \quad \beta^{(l)}(\mathbf{z}) = W_\beta^{(l)}\mathbf{z} + b_\beta^{(l)}. \tag{3}$$

The conditioned feature map of the $l$-th layer is then computed as

$$\tilde{\mathbf{x}}^{(l)} = \gamma^{(l)}(\mathbf{z}) \odot \mathbf{x}^{(l)} + \beta^{(l)}(\mathbf{z}), \tag{4}$$

where $\odot$ denotes channel-wise multiplication, with $\gamma^{(l)}(\mathbf{z})$ and $\beta^{(l)}(\mathbf{z})$ broadcast across the spatial dimensions $(D, H, W)$. $x^l$ is the activation of the $l$-th channel of the input feature map. This formulation allows each U-Net layer to be modulated independently by the metadata, ensuring that subject-specific information influences the hierarchical SVF estimation in a spatially adaptive yet feature-specific manner. The advantages of this approach that the SVF can vary flexibly with respect to $\Delta t$, allowing the model to represent rich, nonlinear, and non-monotonic deformation patterns. Additionally, it does not impose the stationarity assumption across different time horizons.

### 3.2. FutureMorph-S: $\Delta t$-Scaled Integration

In this stationary (FutureMorph-S) framework, the model predicts an SVF independent of $\Delta t$, where the time interval is incorporated only during the integration. This $\Delta_t$-based integration is implemented by adapting SS, effectively scaling the initial velocity field by $\Delta_t$ to approximate the exponential map from 0 to $\Delta_t$ (details in Appendix B). It produces deformation fields that are monotonic with respect to $\Delta t$ by construction. In this case, the SVF network is also simpler, as it does not need to handle temporal variability

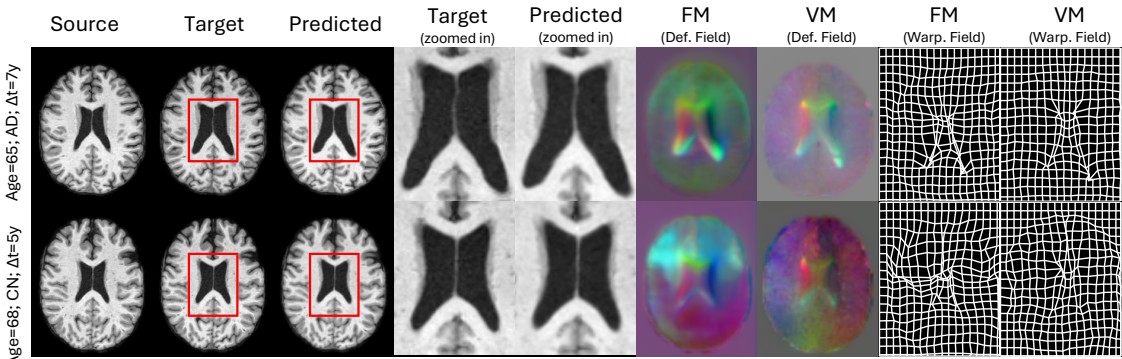

Figure 3: Follow-up predictions for subjects across different ages and time intervals. Left to right: source image, target image, FutureMorph prediction, zoom-in regions, and the FutureMorph deformation field (FM; shown in RGB and grid form), followed by VoxelMorph (denoted as VM). Red boxes highlight substantial ventricular changes. While predicted images visually align with the target in both cases, intensity-based similarity alone does not capture structural discrepancies compared with VoxelMorph.

explicitly. However, a single stationary SVF is too restrictive to capture realistic, heterogeneous anatomical trajectories. Empirically, this yields overly smooth, average deformations, lacking the subject-specific detail or nonlinear progression captured by the model.

## 4. Experiments

**Database and Preprocessing**  We conducted our experiments on the publicly available OASIS-3 dataset (LaMontagne et al., 2019), a large-scale longitudinal neuroimaging dataset containing structural MRI scans of cognitively normal (CN) and diseased (AD) subjects across multiple visits and ages. All images were preprocessed using the Neural Pre-processing (nppy) library (He et al., 2023), including resampling to an isotropic resolution of $[1, 1, 1]$ mm and affine alignment to the MNI512 template. Skull stripping was performed to remove non-brain tissue. In addition, anatomical segmentation was obtained using SynthSeg (Billot et al., 2023), providing cortical and subcortical labels that were later used to evaluate structural consistency in the predicted deformations. The dataset was randomly split into training and test sets at the subject level to prevent data leakage. From these splits, we generated longitudinal pairs for training and evaluation. Specifically, we constructed 844 training and 219 test image pairs $\{I_{t_0}, I_{t_1}\}$, corresponding to two visits per subject ($t_1 > t_0$).

**Implementation Details**  Our model was implemented in PyTorch and trained on a single GPU, NVIDIA-GTX 1080, for 1000 epochs with a batch size of $B = 8$. We used Adam optimizer with a learning rate of $1 \times 10^{-4}$ [2]. The training objective combined an image similarity loss, measured by local normalized cross-correlation (LNCC), with a smoothness

---

2. code available at FutureMorph-Repository

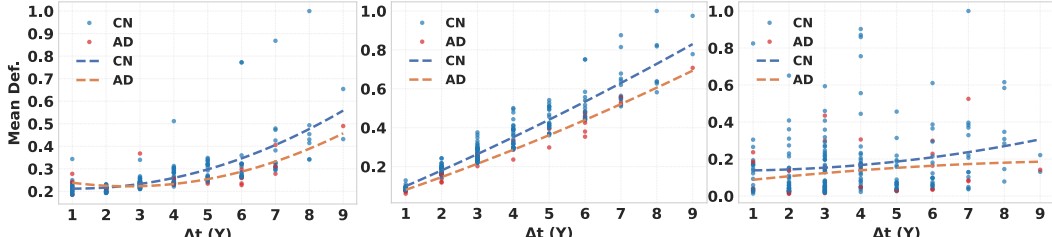

Figure 4: Time-interval trends. Left to right: mean deformation across years for Future-Morph, FutureMorph-S and VoxelMorph. AD and CN refer to the Alzheimer's disease and cognitively normal (control) cohorts. Our model captures growth patterns, showing nonlinear deformation increase with time, unlike linear FutureMorph-S and nearly time-invariant VoxelMorph's pattern.

renegotiation loss, $\mathcal{R}$ mentioned in (2), defined as an $L_1$ gradient penalty, and $\lambda = 0.1$. Categorical variables (e.g., sex and diagnosis) are embedded into continuous vectors and concatenated with continuous features such as time interval $\Delta t$ and age. This unified vector $d_{\text{unified}}$ is passed through a two layers of multilayer perceptron (MLP) with hidden dimension 32, producing the metadata embedding $\mathbf{z} = \mathcal{E}(d_{\text{unified}})$. The embedding $\mathbf{z}$ is then projected through two linear layers to generate the FiLM parameters, $\gamma(\mathbf{z})$ and $\beta(\mathbf{z})$, in Equation (3), which are matched to the channel dimensions of the U-Net layers. These parameters adaptively scale and shift the U-Net feature maps, thereby conditioning the SVF estimation on subject-specific metadata.

**Evaluation** For evaluation, we report MSE, SSIM, and PSNR on the predicted images, and Dice scores on anatomical and Alzheimer's-disease (AD) –related labels. All segmentation maps were obtained using SynthSeg (Billot et al., 2023). We trained VoxelMorph with a two-channel input (fixed and moving images) using the same settings described earlier to ensure a fair comparison. This model serves as a gold-standard upper bound, reflecting the realistic deformation that would be obtained if the follow-up image were available. We further evaluate our predictions by analyzing deformation measures and hippocampal volume changes, and examining their correlations with metadata such as age, time-interval, and relevant subgroup statistics. We compare these analyses with growth trends captured by VoxelMorph and the time-interval–conditioned SS baseline (FutureMorph-S). We analyze FutureMorph's sensitivity to training-data distributions over age and time intervals. We further evaluate sensitivity to each individual metadata variable and to subject-level shuffling of both metadata and baseline anatomy.

## 5. Results and Discussion

Figure 3 shows examples of predicted deformations across two time intervals, age groups, and classes (AD and CN). The first example (top row) aligns well with the VoxelMorph reference, particularly around the ventricles and inferior regions. The second example, however, highlights a limitation: the superior region shows structural discrepancies compared with the VoxelMorph gold standard. Although the predicted follow-up images visually align with the target, intensity-based similarity alone is insufficient. Anatomically plausible defor-

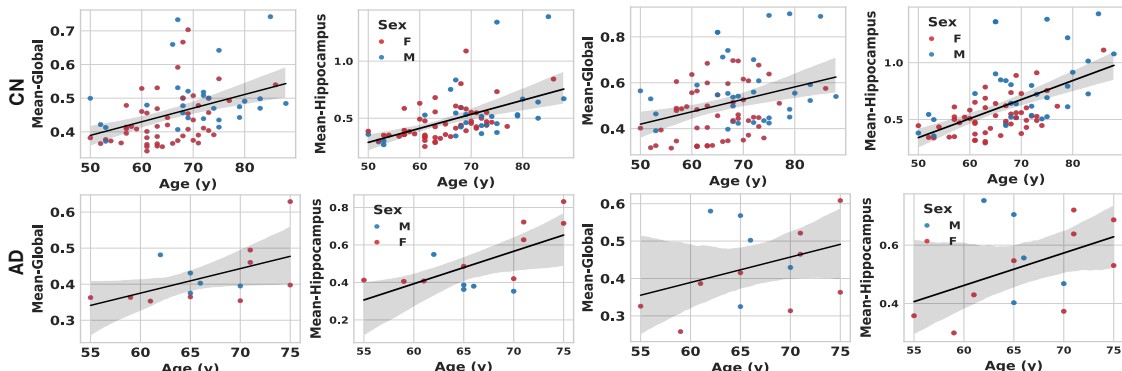

Figure 5: Mean brain-wide and hippocampal deformation versus age for FutureMorph (left) and FutureMorph-S (right) in AD and CN groups, showing consistent age correlations; despite the small AD cohort, clear age-deformation trends are visible, particularly in the larger CN group.

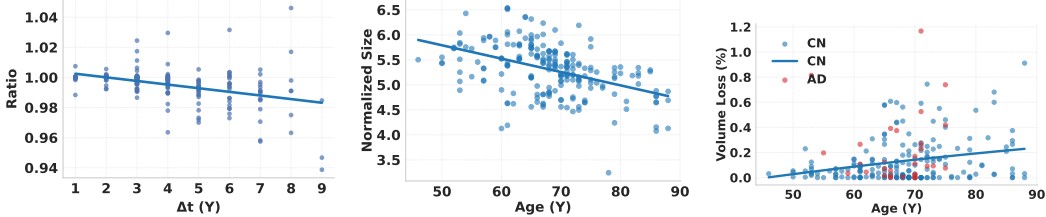

Figure 6: Hippocampal volume trends predicted by FutureMorph. Left to right: volume ratio of predicted follow-up over baseline versus time interval, normalized hippocampal size versus age, and annualized volume loss (%) versus age. The plots show expected age and time-dependent hippocampal changes, with greater loss over longer intervals and increasing annualized loss with age in CN subjects. AD trends are less clear due to smaller sample size. Hippocampal size is the MNI-space volume ($cm^3$, 1mm isotropic) and is shown to illustrate trends rather than absolute native-space values.

mation fields are essential for evaluating the reliability of the follow-up predictions, as they capture subtle and clinically meaningful structural changes. Figure 11, in Appendix D, shows an example of a subject with three visits, illustrating the predicted deformations across longitudinal scans.

**Growth Trends.** Our model successfully learns the underlying growth patterns through the learnable conditioning strategy. As shown in Figure 4, the global deformation magnitude increases with the time interval and follows a nonlinear (approximately quadratic) trend, unlike FutureMorph-S, whose linear behavior is imposed directly by integrating the velocity only up to $\Delta_t$. In contrast, VoxelMorph shows weaker sensitivity to the time interval. Figure 5 further illustrates age-related trends in both AD and CN groups. Although the AD cohort is relatively small, clear correlations between age and deformation measures are visible, especially in the larger CN group. Figure 6 shows hippocampal volume trends predicted by FutureMorph. All hippocampal measures were computed in MNI space and

Table 1: Quantitative comparison against `VoxelMorph` (upper-bound reference). Mean Dice scores are calculated for all and AD-only labels. The vertical line separating the `VoxelMorph` and FutureMorph columns indicates the `VoxelMorph` performance, which serves as a gold-standard upper bound.

|  | VoxelMorph (upper bound) | FutureMorph | FutureMorph-S |
|---|---|---|---|
| MSE ↓ | $0.001 \pm 0.001$ | $0.002 \pm 0.001$ | $0.002 \pm 0.001$ |
| SSIM ↑ | $0.917 \pm 0.072$ | $0.864 \pm 0.082$ | $0.865 \pm 0.081$ |
| PSNR ↑ | $31.729 \pm 3.623$ | $28.598 \pm 3.174$ | $28.548 \pm 3.130$ |
| Dice (All) ↑ | $0.919 \pm 0.039$ | $0.858 \pm 0.096$ | $0.864 \pm 0.084$ |
| Dice (AD) ↑[3] | $0.929 \pm 0.025$ | $0.870 \pm 0.079$ | $0.875 \pm 0.072$ |

therefore represent spatially normalized quantities rather than physical volumes. These measures were used solely to verify longitudinal trends with respect to age and time, which is the primary goal of this analysis. The volume ratio, calculated as $V_{t_1}^{\mathrm{pred}}/V_{t_0}$, decreases with increasing $\Delta_t$, reflecting greater volume loss over longer intervals. Spatially normalized hippocampal size decreases with age, as expected declines with age, as expected. The annualized volume loss, computed as $(V_{t_0} - V_{t_1}^{\mathrm{pred}})/(\Delta_t \cdot V_{t_0}) \times 100$, is small and nearly flat at younger ages in CN subjects, then increases and becomes more variable with age. AD subjects show no clear pattern due to limited sample size, although younger AD subjects exhibit higher annualized loss. These trends are consistent with expected anatomical changes and with findings from longitudinal studies, demonstrating FutureMorph's ability to model age- and time-dependent hippocampal changes.

**Evaluation Metrics and Comparison** Table 1 reports quantitative metrics (SSIM, PSNR, MSE) and mean Dice score across all anatomical and AD-related labels for FutureMorph and FutureMorph-S, compared with the VoxelMorph upper-bound reference. Paired t-test results indicate that FutureMorph differs significantly from VoxelMorph across all evaluated metrics. In contrast, FutureMorph and FutureMorph-SS show a significant difference only in Dice score, while no statistically significant differences are observed for the remaining metrics. Detailed p-values from the paired t-tests are reported in Table 2, Appendix D. While FutureMorph and FutureMorph-S achieve similar intensity-based metrics, FutureMorph-S is limited in capturing structural changes, as illustrated in Figure 7. These results support our claim that predicting future deformation provides more meaningful anatomical insight than relying solely on global intensity measures.

**Limitations of FutureMorph-S.** Figure 7 illustrates the limitations of FutureMorph-S. In this experiment, we generate future deformations for each test subject at multiple time intervals to examine model behavior for longer follow-up periods. While FutureMorph-S shows increasing global mean deformation with longer time intervals—owing to its direct scaling of the stationary velocity field—it fails to capture structural details and produces less realistic deformations as the interval grows (as shown in the left side of Figure 7). In contrast, FutureMorph produces realistic, anatomically plausible deformations without a strict age-dependent monotonic trend, reflecting that factors other than age, such as

---

3. AD-relevant labels: hippocampus, amygdala, thalamus, cerebral white matter, and cerebral cortical gray matter.

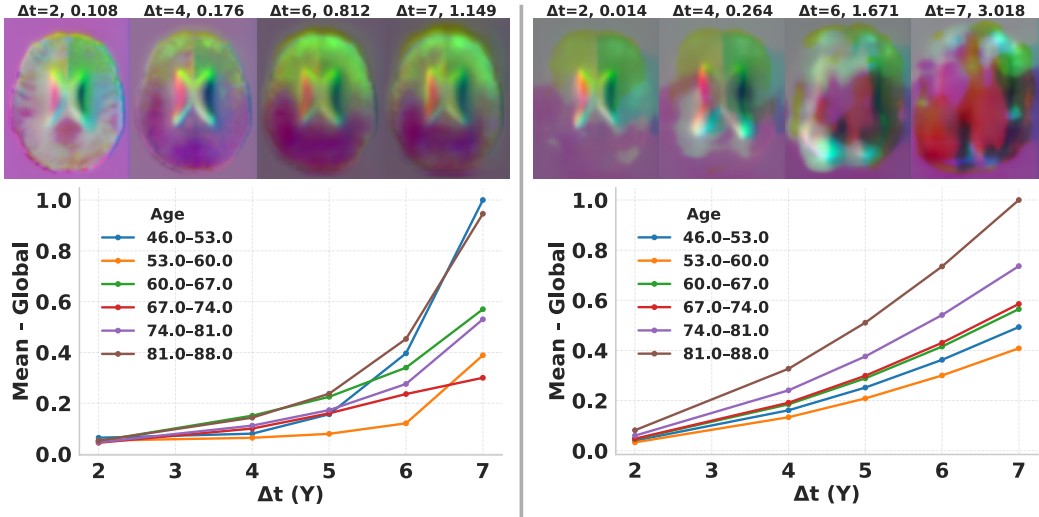

Figure 7: Limitation of FutureMorph-S. Predicted deformation fields for different time intervals for the same subject (top) and global mean deformation across subjects for each group over time intervals, comparing FutureMorph (left) and FutureMorph-S (right). FutureMorph-S, although reflecting increased growth with age and time interval, fails to generate realistic deformations at larger intervals and lacks structural details.

disease status (AD vs CN), influence global brain changes. FutureMorph-S fails because the stationary SVF assumption forces it to learn a single "average" deformation, preventing it from capturing subject-specific structural changes. A more detailed analysis, including examples of these failures, is provided in C.

**Sensitivity to Training Data Distributions** We examined how sensitive FutureMorph is to the distributions of age and time intervals observed during training. Test cases were grouped into age bins (2-year intervals and decade ranges) and time-interval bins, and Dice scores were computed per group alongside the corresponding training-set counts. As shown in Figure 8, the performance degraded noticeably for time-interval ranges that were underrepresented in the training data, whereas age distributions had a weaker effect. This indicates that time interval between visits is a dominant factor for prediction accuracy, consistent with our modeling assumptions.

**Sensitivity to Shuffling** We performed an ablation study to evaluate the contribution of individual metadata variables by shuffling diagnosis, age, or time interval separately. Table 3 in Appendix D reports Dice scores for each case, showing larger drops for age and time interval than for diagnosis. These results indicate that these factors are particularly important for conditioning and support the claim that FutureMorph captures meaningful subject-specific effects beyond temporal spacing alone. To evaluate the sensitivity of FutureMorph's conditioning and verify that predictions are subject-specific rather than driven by population priors, we performed controlled shuffling of metadata and baseline anatomy. For metadata shuffling, we permuted the conditioning variables while keeping the anatomy fixed. For anatomy shuffling, we replaced the moving image with that of another subject matched for age and disease class while keeping the segmentation unchanged. Metadata

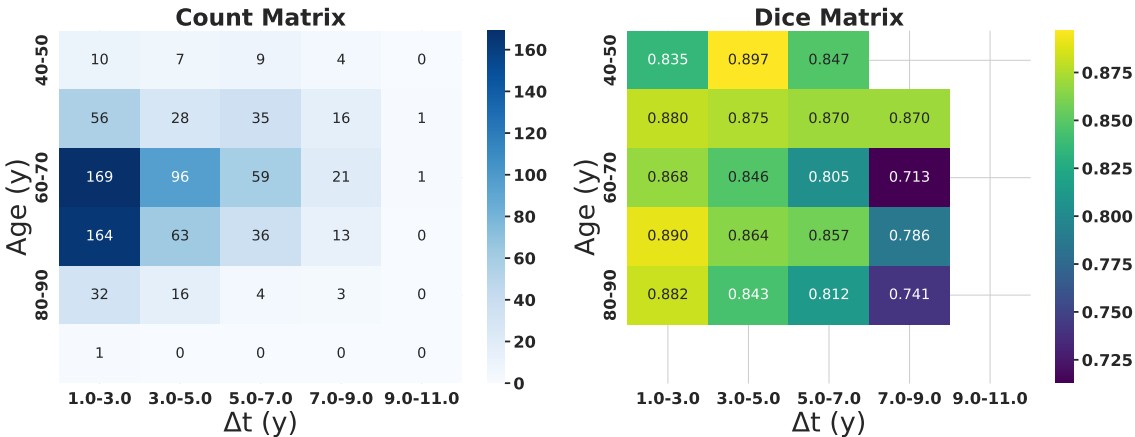

Figure 8: Sensitivity to training data distribution. Left: count matrix of training samples across age and time-interval bins. Right: mean Dice scores of test cases computed over the same bins. Performance drops in sparsely populated time-interval ranges indicate that prediction accuracy is more sensitive to temporal spacing than to age distribution.

permutation substantially degraded Dice (See Table 4, Appendix D), whereas anatomy shuffling primarily affected intensity-based metrics. The anatomy shuffle produced the largest overall degradation with milder effects on Dice, indicating that metadata mainly governs anatomical consistency while baseline anatomy drives intensity fidelity. These results indicate that FutureMorph leverages both subject-specific anatomy and metadata conditioning rather than population-average priors.

**Consistency with VoxelMorph.**

We compare FutureMorph and VoxelMorph using the mean Jacobian determinant within hippocampal regions, verifying that the predicted forecasting deformations are consistent with standard registration fields (Fig. 9). In addition, group-wise differences in mean hippocampal deformation (AD vs. CN and female vs. male) are consistent with registration-based measurements, indicating that the predicted deformations capture meaningful anatomical patterns. Although hippocampal volumes should ideally be computed in native subject space and corrected for fixed effects and intracranial volume, we report measurements in MNI space solely to assess agreement with registration-based deformations. We emphasize that these MNI-space measurements are not intended to represent physical hippocampal volumes and should not be used for clinical interpretation.

## 6. Conclusion

We presented a framework for predicting future diffeomorphic deformations from baseline MRI and metadata, enabling generating subject-specific follow-up and capturing growth and aging trends. This problem can also be viewed as registration with a missing fixed image. Our conditional mechanism evaluates how well future deformation candidates, estimated from metadata, match the deformation reference obtained if the follow-up image were available. While the time interval is the most informative factor, the conditioning

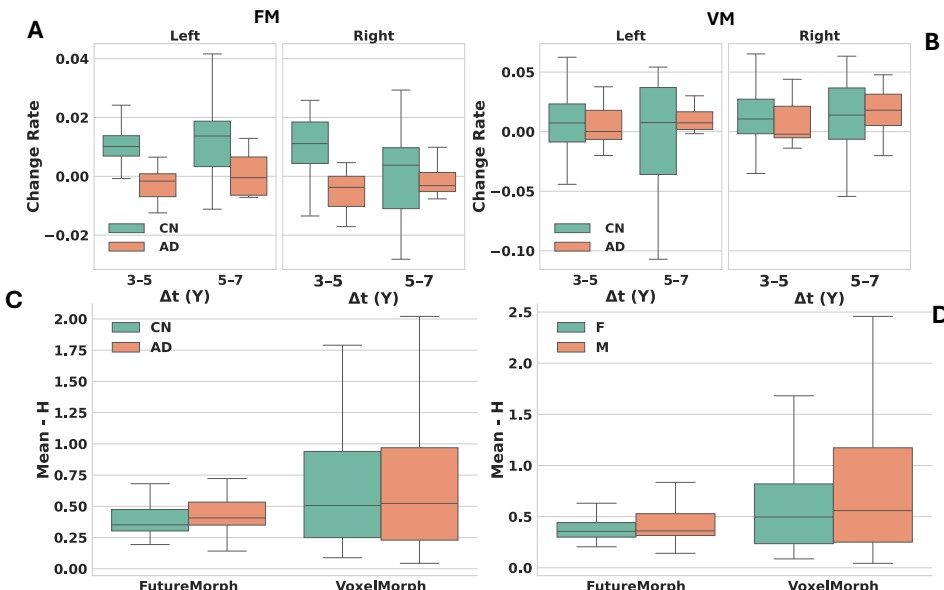

Figure 9: (A–B) Mean Jacobian determinant within the left and right hippocampus, predicted by FutureMorph (A) and VoxelMorph (B) for CN and AD groups. (C) Mean deformation magnitude for CN and AD. (D) Mean deformation magnitude for female (F) and male (M) subjects across both models. Paired comparisons of the mean hippocampal Jacobian determinant between FutureMorph and VoxelMorph yielded p-values greater than 0.05, indicating no statistically significant differences between the two methods.

strategy strongly influences deformation quality and structural accuracy. Shuffling experiments show that metadata conditioning is critical for anatomical accuracy, while baseline anatomy primarily governs intensity fidelity, confirming that FutureMorph relies on both rather than population-average priors.

We also show that learnable conditioning outperforms simple integration-based approaches in the scaling-and-squaring module, which fail due to the stationary velocity assumption. While our approach provides clinically meaningful predictions, it is limited by the reliance on available training data and potential bias in underrepresented subgroups (e.g., AD subjects). Incorporating uncertainty estimation would be crucial to quantify confidence in predictions and support safe clinical use. The small AD cohort is a limitation and contributes to dataset imbalance; this could be mitigated in future work using targeted sampling strategies during training.

Since there is no direct way to validate our predictions, we used VoxelMorph as a reference to assess agreement of FutureMorph forecasts with registration-based deformations. Future work could incorporate classifier-based guidance on predicted future images or volume-based loss to further support the biological plausibility of the forecasts. Another promising direction is to incorporate an auxiliary network that predicts the time interval from the predicted deformation fields, which could regularize training and encourage temporally consistent and biologically realistic deformations.

## Acknowledgments

S.K is a fellow of the Eric and Wendy Schmidt Postdoctoral Award for Women in Mathematical and Computing Sciences (this research received support through Schmidt Sciences). The authors gratefully acknowledge the funding provided by the Sidney and Vivian Konigsberg Lectureship at Jacobs Technion-Cornell Institute. S.K would like to thank the VATAT fellowship for the funding support.

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

---

**Algorithm 1** Time-Conditioned SS

---

**Input:** Stationary SVF $V$, time interval $\Delta t$, squaring steps $N$
**Output:** Integrated deformation $\phi^{\Delta t}$
$h \leftarrow \dfrac{\Delta t}{2^N}$ ;                                    // integration step size
$u \leftarrow h \cdot V$ ;                                    // initial small displacement
**for** $k \leftarrow 1$ **to** $N$ **do**
$\quad \mid \quad u \leftarrow u + \text{Warp}(u, u)$ ;                                    // composition via warping
**end**
**return** $\phi^{\Delta t}(x) = x + u(x)$

---

## Appendix A. Details on the Architecture of the U-Net

Our backbone comprises an encoder and decoder with skip connections. Both encoder and decoder parts consist of CNN layers with kernel size $3 \times 3 \times 3$, followed by Leaky ReLU activation functions. The encoder consists of successive 3D convolutional layers with channel configuration $[16, 32, 32, 32]$, while the decoder consists of 3D deconvolutional layers with channels $[32, 32, 32, 32, 16, 16]$. A final 3-channel convolutional layer maps the decoder output to the SVF.

## Appendix B. FutureMorph-S: Conditioned Diffeomorphic Integration via SS

Diffeomorphic deformation is defined as the flow $\{\phi^t\}_{t \geq 0}$ solving the equation (1), where $V$ is non-stationary VVF. The solution at time $t$ is the Lie exponential of $tV$:

$$\phi^t = \exp(t\,V). \tag{5}$$

We use the SS strategy to approximate the integration to an arbitrary time interval $\Delta t$, by calculation of the following exponential term:

$$\exp(\Delta t\,V) = \left[\exp\left(\frac{\Delta t}{2^N}v\right)\right]^{2^N} = \underbrace{\phi^h \circ \cdots \circ \phi^h}_{N \text{ compositions}}, \qquad h = \frac{\Delta t}{2^N}. \tag{6}$$

Thus, scaling the SVF by $\Delta t$ is equivalent to solving the integration in (1) over the horizon $[0, \Delta t]$.

We approximate the small-step flow by a first-order update:

$$\phi^h(x) \approx x + h\,v(x), \qquad h = \frac{\Delta t}{2^N}. \tag{7}$$

Repeated composition via SS then yields the full diffeomorphic deformation field, $\phi^{\Delta t}$, as outlined in algorithm 1. This procedure is numerically stable when initial displacements are small (i.e., large enough $N$), and preserves diffeomorphic invertibility of the solution.

## Appendix C. On the Limitation of FutureMorph-S for Non-Stationary SVFs

As we discussed earlier, the adaptive SS integration scheme assumes that the deformation evolves from a stationary SVF $V(x)$, such that the deformation over a time interval $\Delta t$ is given by (5).

This assumption holds only when $V(x)$ is stationary and does not change with time. In the general non-stationary case $V(t, x)$, the true flow, defined in (1), has the following analytic solution:

$$\phi_{\Delta t}(x) = x + \int_0^{\Delta t} V\big(t, \phi_t(x)\big) \, dt \tag{8}$$

The SS formulation, which simply scales $V(x)$ by $\Delta t$ and exponentiates, ignores temporal variation in $V(t, x)$ and therefore fails to represent non-stationary dynamics.

**Claim 1** *Let $V(t, x) = c(t)\, x$ be a one-dimensional, spatially linear SVF. The exact flow is*

$$\phi_{\Delta t}(x) = \exp\left( \int_0^{\Delta t} c(s) \, ds \right) x,$$

*while the SS approximation gives*

$$\hat{\phi}_{\Delta t}(x) = \exp\big(c(0)\, \Delta t\big) x.$$

*These two expressions are equal for all $\Delta t$ if and only if*

$$\int_0^{\Delta t} c(s) \, ds = c(0)\, \Delta t \iff c(t) \text{ is constant on } [0, \Delta t].$$

Here we show toy examples that illustrate the failure of SS in non-stationary case.

### 1D Example

Consider $c(t) = \lambda_0 + kt$, representing a linearly evolving growth rate. The true flow is

$$\phi_{\Delta t}(x) = \exp\big(\lambda_0 \Delta t + \tfrac{1}{2}k\Delta t^2\big) x,$$

while the stationary approximation yields $\hat{\phi}_{\Delta t}(x) = \exp((\lambda_0 + k \cdot 0)\Delta t)x = e^{\lambda_0 \Delta t}x$. The deviation grows quadratically with $\Delta t$ and with the non-stationarity parameter $k$, illustrating that adaptive SS enforces an artificially constant deformation rate.

### 2D Example

For the two-dimensional example $V(t, \mathbf{x}) = c(t)\, \mathbf{x}$, the true deformation is

$$\phi_{\Delta t}(\mathbf{x}) = \exp\left( \int_0^{\Delta t} c(s) \, ds \right) \mathbf{x},$$

whereas the stationary SS approximation replaces the integral by $c(0)\Delta t$, yielding $\exp(c(0)\Delta t)\mathbf{x}$. As illustrated in Figure 10, this causes the stationary prediction to diverge rapidly from

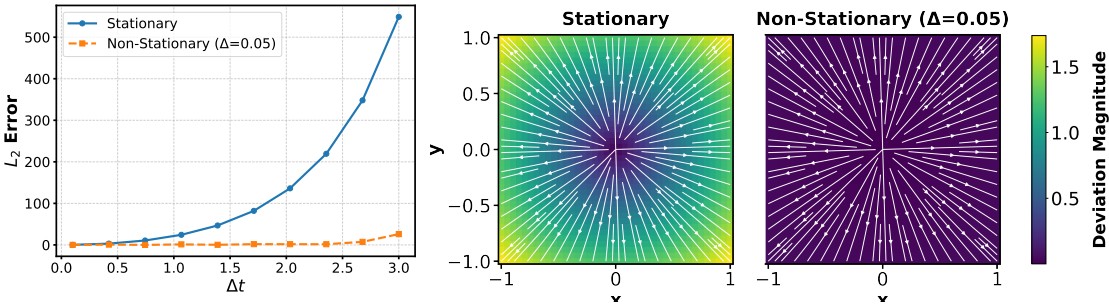

Figure 10: Comparison between stationary and non-stationary SS. Left: The error between predicted and analytic deformations grows rapidly with $\Delta t$ under the stationary assumption. Right: Deviation maps at $\Delta t = 2$ show that the standard SS overestimates expansion, whereas the proposed time-sliced non-stationary SS closely matches the ground truth.

the analytic solution as $\Delta t$ increases, producing uniformly scaled and overly smooth fields. To demonstrate this, we compute a non-stationary SS by dividing $[0, \Delta t]$ into small steps of size $\delta = 0.05$ and composing incremental updates using the midpoint velocity in each interval. This time-sliced integration closely follows the ground truth and highlights the failure of the stationary assumption.

## Appendix D. Extended Analysis and Ablations

Table 2: P-values of paired t-tests comparing the proposed method `FutureMorph` (FM) with `VoxelMorph` (VM) and `FutureMorph-S` (FM-S) across evaluation metrics.

| Metric | FM vs. VM | FM vs. FM-S |
|--------|-----------|-------------|
| MSE | $2.27 \times 10^{-4}$ | 0.575 |
| SSIM | $3.77 \times 10^{-5}$ | 0.114 |
| Dice | $9.53 \times 10^{-4}$ | $3.37 \times 10^{-3}$ |

Table 3: Sensitivity of FutureMorph to shuffling of individual metadata variables. Performance is reported as mean $\pm$ standard deviation (Dice). Metadata shuffling was performed for diagnosis (class), age, and time interval ($\Delta_t$) separately.

| | Class (diagnosis) | Age | $\Delta_t$ | FutureMorph |
|--|-------------------|-----|-----------|-------------|
| Dice (All) ↑ | $0.730 \pm 0.165$ | $0.618 \pm 0.161$ | $0.620 \pm 0.159$ | $0.858 \pm 0.096$ |
| Dice (AD) ↑ | $0.726 \pm 0.148$ | $0.640 \pm 0.115$ | $0.641 \pm 0.114$ | $0.870 \pm 0.079$ |

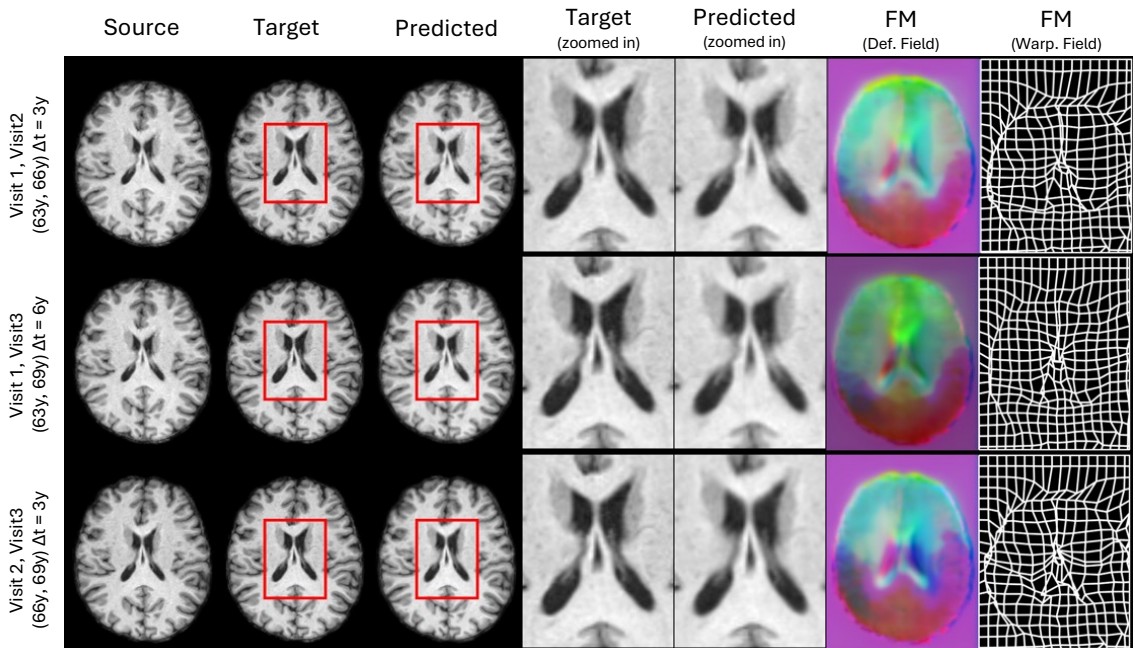

Figure 11: Follow-up predictions for the same subject across different visits. Left to right: source image, target image, FutureMorph prediction, zoom-in regions, and the FutureMorph deformation field (FM; shown in RGB and grid form). Red boxes highlight substantial ventricular changes. The age and time intervals are highlighted in the left side of each row.

Table 4: Sensitivity of FutureMorph to metadata (M-Shuffle) and anatomical (A-Shuffle) shuffling. Performance is reported as mean ± standard deviation. Anatomy shuffling was conducted by replacing the moving image with that of another subject matched for age and disease class while keeping the segmentation unchanged. Performance degradation under shuffling indicates that metadata primarily affects anatomical accuracy (Dice), whereas baseline anatomy governs intensity-based metrics, confirming that FutureMorph relies on both rather than population-average priors.

| | M-Shuffle | A-Shuffle (Match) | FutureMorph |
|---|---|---|---|
| MSE ↓ | $0.002 \pm 0.001$ | $0.007 \pm 0.002$ | $0.002 \pm 0.001$ |
| SSIM ↑ | $0.862 \pm 0.082$ | $0.688 \pm 0.082$ | $0.864 \pm 0.082$ |
| PSNR ↑ | $28.392 \pm 3.051$ | $21.747 \pm 1.747$ | $28.598 \pm 3.174$ |
| Dice (All) ↑ | $0.628 \pm 0.163$ | $0.854 \pm 0.096$ | $0.858 \pm 0.096$ |
| Dice (AD) ↑ | $0.647 \pm 0.122$ | $0.867 \pm 0.078$ | $0.870 \pm 0.079$ |

