# OpenReview forum: "FutureMorph: Toward Predicting Future Deformation Fields in Longitudinal Imaging"
_MIDL.io/2026/Conference — MIDL 2026 Poster_

### Official Review · Reviewer_oAvn · 2025-12-29

**Confidence:** 4
**Preliminary Rating:** 4
**Final Rating:** 5

**Summary:**

The paper defines a new registration problem as: given an image at $t_0$ and auxiliary data (age, sex, etc.), can we make a plausible deformation mapping from $t_0$ to a later time $t_1$. The authors solve the question by proposing $\textbf{FutureMorph}$, which uses a conditional UNet to generate the deformation field and the spatial transformation field to synthesize image. Experiments on OASIS-3 dataset show that FutureMorph captures realistic aging- and disease-related anatomical trajectories, including nonlinear growth trends and hippocampal volume loss patterns, and produces more anatomically plausible deformations than the stationary baseline, especially at longer time horizons.

**Strengths:**

1. The reformulation of longitudinal prediction as forecasting future diffeomorphic deformation fields rather than synthesizing future images is particularly valuable because deformation fields are anatomically interpretable and directly comparable to classical morphometric analyses.
2. The evaluation goes beyond image similarity metrics by demonstrating correlations between deformation-derived measures, including global deformation magnitude, age-related trends, and hippocampal volume loss, and subject-level metadata. These relationships align with established biological knowledge of aging and neurodegeneration.

**Weaknesses:**

1. The experimental validation is limited in scope, as all results are reported on a single dataset and the training and evaluation setup relies exclusively on patients with two visited. Patients with more than two visits are not considered.
2. The AD cohort is relatively small, making the correlation between mean deformation vs. age weak (Figure 5, bottom).

**Detailed Comments:**

1. It would be interesting to see the sensitivity of the method to subject-level metadata. In particular, evaluating performance when baseline images are paired with incorrect or shuffled metadata at test time can show how strongly the predicted deformation depends on each metadata component.

2. As a potential direction for future work, the authors might consider using an auxiliary mapping, for example a network $D$ that predicts time interval $\Delta t$ from a deformation field $\Phi$, during the training. It may provide an additional self-supervisory signal or regularization mechanism for longitudinal modeling.

**Justification Of Final Rating:**

Thank the authors for addressing my concern. I appreciate the additional experiment on studying the sensitivity to the metadata, and a visualization on subjects with multiple visits provides more insights to the approach. I have raised my score from 4 to 5. Good work!

**Justification Of The Preliminary Rating:**

Despite being evaluated on a single dataset, the paper addresses an interesting and clinically relevant task that is well motivated. The results demonstrate that the proposed method is effective and capable of generating realistic deformation fields that capture meaningful biological signals, such as age- and disease-related anatomical changes.

**Questions To Address In The Rebuttal:**

The paper would be strengthened by clarifying whether additional samples from the Alzheimer’s disease group are available, or by discussing how the small AD cohort affects the reliability of the reported disease-specific trends. In addition, providing at least one visual example of a patient with multiple visits, even if the model is still trained on paired data, would help illustrate how the predicted deformations behave across successive time intervals.

---

> ### Author Response · Authors · 2026-01-23
> **Response to Reviewer oAvn (1/2)**
>
> We thank the reviewer for the interesting comments and insightful suggestions. (All modifications are highlighted in red in the revised manuscript.)
>
> **W1:** The experimental validation is limited in scope, as all results are reported on a single dataset and the training and evaluation setup relies exclusively on patients with two visits. Patients with more than two visits are not considered.
>
> **W2:** The AD cohort is relatively small, making the correlation between mean deformation vs. age weak (Figure 5, bottom).
>
> **A1 + A2:** We thank the reviewer for the general comments. Our experiments focus on the OASIS-3 dataset, where relatively few subjects have more than two visits; this motivated our formulation based on scan pairs, yielding 814 training and 219 test pairs. While this setting allows us to introduce the forecasting problem and demonstrate the benefits of conditioning, our framework naturally extends to multi-visit scenarios because we construct multiple visit combinations for each subject when available. We include illustrative multi-visit results and the corresponding predicted deformations in **Figure 11, Appendix D**. Extending the approach to longer trajectories and additional datasets remains an important direction for future work.
>
> We agree that the AD cohort is relatively small, which weakens correlations between deformation measures and age (Fig. 5, bottom). This limitation is now explicitly noted in the figure caption, in the “Growth Trends” discussion, and in the Conclusion.
>
> **DC1:** It would be interesting to see the sensitivity of the method to subject-level metadata. In particular, evaluating performance when baseline images are paired with incorrect or shuffled metadata at test time can show how strongly the predicted deformation depends on each metadata component.
>
> **A1:** We thank the reviewer for pointing out this interesting point. We added an ablation study in which metadata are shuffled at test time to assess sensitivity to individual conditioning variables. This analysis is described in the new **Sensitivity to Shuffling** paragraph in the Results and Discussion section, with quantitative results reported in **Table 4 (first column), Appendix D**. The observed degradation in Dice and image-based metrics demonstrates that FutureMorph meaningfully leverages subject-level metadata in addition to baseline anatomy. In addition, we performed an ablation study to evaluate the contribution of individual metadata variables by shuffling diagnosis, age, or time interval separately. **Table 3 in Appendix D** reports Dice scores for each case, showing larger drops for age and time interval than for diagnosis. These results show that these factors are important for conditioning and that FutureMorph captures meaningful subject-specific effects beyond time interval alone.
>
> **DC2:** As a potential direction for future work, the authors might consider using an auxiliary mapping, for example a network  that predicts time interval  from a deformation field , during the training. It may provide an additional self-supervisory signal or regularization mechanism for longitudinal modeling.
>
> **A2:** We thank the reviewer for this insightful suggestion. We have addressed this potential future direction in the revised manuscript. We mentioned in the conclusion: *“Another promising direction is to incorporate an auxiliary network that predicts the time interval from the predicted deformation fields, which could regularize training and encourage temporally consistent and biologically realistic deformations.”*

---

> ### Author Response · Authors · 2026-01-23
> **Response to Reviewer oAvn (2/2)**
>
> **Question to address**
>
> **Q1:** The paper would be strengthened by clarifying whether additional samples from the Alzheimer’s disease group are available, or by discussing how the small AD cohort affects the reliability of the reported disease-specific trends. In addition, providing at least one visual example of a patient with multiple visits, even if the model is still trained on paired data, would help illustrate how the predicted deformations behave across successive time intervals.
>
> **A1:** We thank the reviewer for the valuable comments. While only a few subjects have more than two visits—which is the main reason we focused on pairs—we provide a visual example of deformation for two consecutive follow-ups; **Figure 11 in  Appendix D**. Regarding the small AD cohort and its effect on the reliability of disease-specific trends, we acknowledge that fewer samples make it more challenging to infer correlations with age and time interval. Nevertheless, even within the AD group, we are able to capture the expected trend of hippocampal volume change with age, and the predicted deformations remain higher in AD than in CN subjects despite the smaller sample size (see Figure 8-C). These observations are consistent with VoxelMorph outcomes, supporting the validity of the reported trends. In addition, we discussed this limitation and how it can be tackled in future work in the conclusion: *“The small AD cohort is a limitation and contributes to dataset imbalance; this could be mitigated in future work using targeted sampling strategies during training.”*

---

### Official Review · Reviewer_X9Jb · 2026-01-05

**Confidence:** 3
**Preliminary Rating:** 4
**Final Rating:** 4

**Summary:**

This paper presents a framework for predicting subject-specific deformation fields using only a baseline image and subject meta data. The model, called FutureMorph, predicts a stationary velocity field that is then integrated to yield a diffeomorphism. These deformation fields can then be used to synthesize subsequent timepoints to generate longitudinal data. The authors evaluate their model using the OASIS-3 dataset, and show that the resulting transformations can capture subject-specific, anatomical changes similar to VoxelMorph.

**Strengths:**

- The concept of synthesizing a deformation (and subsequent, longitudinal data) is an interesting problem.
- The implementation of an SVF-only version of the model was a clever way to demonstrate the importance of meta-data conditioning within the framework.
- The literature review presents a sufficient background, and the overarching research question is clearly and succinctly stated in the introduction.

**Weaknesses:**

- I am not sure that the magnitude of the predicted vs. target deformation fields is a relevant metric. Although it is useful for assessing their similarity, it seems like the Dice/MSE/volumes are more clinically relevant for downstream tasks. I think the paper would be stronger if the emphasis were placed on these metrics instead of on the deformation magnitude (mean - global).
- The paper is difficult to read at times, primarily due to lengthy sentences. There were also a few grammatical errors throughout the text.
- There is no discussion of the statistical significance of any results.
- The authors report a small sample size with respect to AD data (although they do list this as a limitation to their method).

**Detailed Comments:**

- Figure 3: include the abbreviations for FutureMorph (FM) and VoxelMorph (VM) within the caption, since they are used in the figure.
- Figure 4: the vertical axis has no label, making this figure difficult to intepret.
- Figure 6: are the reported volumes in subject- or MNI152-space? If in MNI, it would be better to convert back to subject. Volumes should also be corrected for fixed effects and for estimated intracranial volume.
Table 1: why is the only vertical line between the VoxelMorph and FutureMorph columns? Also, consider rounding to 2 or 3 decimal points for brevity...
Figure 8: It is not clear which panel (A or B) corresponds to FutureMorph and which corresponds to VoxelMorph. This should be clarified in the caption and/or in the figure subplot titles.
Figures (general): the paper would look more consistent if Figures 4-7 had been configured to be relatively the same size. As it is, Figure 7 is far easier to read than the others.

**Justification Of Final Rating:**

The authors addressed all the concerns that I had (lack of statistical analysis, grammatical errors, and some minor issues with their Figures). They also answered the questions I had regarding their evaluation metrics (particularly the mean-global deformation field). I have updated my review to a 4 instead of a 3.

**Justification Of The Preliminary Rating:**

I rated this paper as "borderline" because although I think the problem/methodology is interesting and thoroughly investigated, some of the analysis of results could be reformulated to better highlight the downstream, clinical application of this framework. This could be achieved through the suggestions I made in the detailed comments. I also indicated in that section a few others questions/concerns I had which lead to this rating.

**Questions To Address In The Rebuttal:**

- Please include a statistical analysis of the results, particularly those depicted in Table 1 and Figure 8. It is difficult to interpret the results in Figure 8 because I am unsure which of the top panels (A/B) corresponds to FutureMorph and which to VoxelMorph, and also because it is not clear whether the data exhibits any significant differences (e.g., between volume differences calculated using results from each method).
- Is mean - global deformation magnitude a common metric when assessing the similarity of deformation fields? I think it would be better to focus on the Dice/MSE between resulting images/labels synthesized using these transformations, as this would better highlight the method's clinical applications. The abstract highlights that the method produces "clinically meaningful predicted deformations", but I am not sure if deformation magnitude is aa clinically meaningful of a metric compared to others.

---

> ### Author Response · Authors · 2026-01-23
> **Response to Reviewer X9Jb (1/2)**
>
> We appreciate the reviewer’s insightful feedback and address the general comments below. (All modifications are highlighted in red in the revised manuscript.)
>
> **Weaknesses:**
>
> **A:**
> - The magnitude of predicted deformation fields (global and hippocampal) is reported to verify trends and correlations with variables such as age and time interval. Clinically relevant metrics—Dice, SSIM, and MSE—are computed on anatomical segmentations and predicted images to assess prediction accuracy against ground truth.
> - We thank the reviewer for this comment. We revised the manuscript for wordy or lengthy sentences, splitting them to improve readability, and corrected typos and grammar errors. For example, in the Introduction, “though” was changed to “although,” and the sentence starting with “Our perspective provides a mechanism both for understanding future geometric changes…” was revised for conciseness.
> - We have added discussion of statistical significance for Table 1 and Figure 8 (Figure 9 in the modified version). ( Table 2 in appendix D presents p-values for Table 1 and caption in Figure 9 summarizes the statistical test results)
> - We acknowledge that the small AD cohort is a limitation, contributing to dataset imbalance; this can be mitigated in future work via sampling strategies during training, as noted in the revised **Conclusion**: *“The small AD cohort is a limitation and contributes to dataset imbalance; this could be mitigated in future work using targeted sampling strategies during training.”*
>
>
>
> **Detailed Comments:**
> - Figure 3: include the abbreviations for FutureMorph (FM) and VoxelMorph (VM) within the caption, since they are used in the figure.
> - Figure 4: the vertical axis has no label, making this figure difficult to interpret.
> - Figure 6: are the reported volumes in subject- or MNI152-space? If in MNI, it would be better to convert back to subject.
> - Volumes should also be corrected for fixed effects and for estimated intracranial volume. Table 1: why is the only vertical line between the VoxelMorph and FutureMorph columns? Also, consider rounding to 2 or 3 decimal points for brevity... Figure 8: It is not clear which panel (A or B) corresponds to FutureMorph and which corresponds to VoxelMorph. This should be clarified in the caption and/or in the figure subplot titles. Figures (general): the paper would look more consistent if Figures 4-7 had been configured to be relatively the same size. As it is, Figure 7 is far easier to read than the others.
>
>
> **A**:
> - We modified the caption of Figure 3 to include the abbreviations for FutureMorph (FM) and VoxelMorph (VM) as suggested.
> - Vertical axis (y-axis) label were added to Figure 4
> - In the revised paper all figures size were configured to be relatively the same size.
> - We thank the reviewer for this important observation. We agree that physical volumes should ideally be calculated in subject space and corrected for intracranial volume and other covariates. In our current analysis, hippocampal measures were computed in MNI space and used solely to capture longitudinal trends and patterns over age, time, or disease, rather than absolute volumetry. Figure 6 labels and the caption have been updated accordingly and this clarification has been added to the Results and Discussion (**Growth Trends** paragraph): *”All hippocampal measures were computed in MNI space and therefore represent spatially normalized quantities rather than physical volumes. These measures were used solely to verify longitudinal trends with respect to age and time, which is the primary goal of this analysis”*.
> Correspondingly, Figure 9 (formerly Figure 8) now reports regional hippocampal Jacobian determinants instead of volume loss. Subject-space volumetric analysis remains an important direction for future work.
> Table 1 was revised to round all values to three decimal places for consistency, and the caption now clarifies that the vertical line denotes VoxelMorph as the gold-standard upper-bound reference.
> FutureMorph (FM) and VoxelMorph (VM) abbreviations were added to Figure 9 (A-B) (formerly Figure 8), and p-values assessing statistical significance between FM and VM were computed and reported in the figure caption.

---

> ### Author Response · Authors · 2026-01-23
> **Response to Reviewer X9Jb (2/2)**
>
> **Questions to address:**
>
> **Q1:** Please include a statistical analysis of the results, particularly those depicted in Table 1 and Figure 8. It is difficult to interpret the results in Figure 8 because I am unsure which of the top panels (A/B) corresponds to FutureMorph and which to VoxelMorph
>
> **A1:** We thank the reviewer for this comment. **Figure 9 (formerly Figure 8)** has been updated to clearly label FutureMorph (FM) and VoxelMorph (VM) in panels A and B, and p-values assessing statistical significance between FM and VM are now reported in the caption. For **Table 1**, we computed paired t-test p-values for all three methods and reported them in **Table 2 (Appendix D)**. These updates are described in the Evaluation Metrics and Comparison paragraph: *“Paired t-test results indicate that FutureMorph differs significantly from VoxelMorph across all evaluated metrics. In contrast, FutureMorph and FutureMorph-SS show a significant difference only in Dice score, while no statistically significant differences are observed for the remaining metrics. Detailed p-values from the paired t-tests are reported in Table 2, Appendix D.”*
>
>
> **Q2:** Is mean - global deformation magnitude a common metric when assessing the similarity of deformation fields? I think it would be better to focus on the Dice/MSE between resulting images/labels synthesized using these transformations, as this would better highlight the method's clinical applications. The abstract highlights that the method produces "clinically meaningful predicted deformations", but I am not sure if deformation magnitude is a clinically meaningful metric compared to others.
>
> **A2:** We thank the reviewer for pointing out this matter. Global deformation magnitude, as well as hippocampus-specific magnitude, is reported to illustrate structural change trends across time and assess correlations with age and time interval. While metrics such as Dice, MSE, and SSIM capture overall image similarity, they do not directly convey the extent or direction of anatomical growth or atrophy. Deformation magnitude provides a complementary, interpretable measure of structural change, highlighting trends that are clinically meaningful beyond conventional similarity metrics. To clarify this distinction, the **revised manuscript** separates deformation-based analyses from prediction-performance metrics into two dedicated paragraphs: **Evaluation Metrics and Comparison**, which focuses on Dice/MSE/SSIM, and **Growth Trends**, which analyzes deformation magnitudes and their relationships with age and time.”

---

> > ### Comment · Reviewer_X9Jb · 2026-01-26
> >
> > The authors addressed all the concerns that I had, and I have updated my review to a 4 instead of a 3.

---

### Official Review · Reviewer_R7UD · 2026-01-17

**Confidence:** 3
**Preliminary Rating:** 5
**Final Rating:** 5

**Summary:**

This paper introduces FutureMorph, a framework for predicting future diffeomorphic deformation fields in longitudinal MRI using only a baseline scan and subject level metadata such as age, sex, diagnosis, and time interval. The key idea is to reframe longitudinal forecasting as “registration with a missing fixed image”, where the model learns to predict stationary velocity fields conditioned on metadata to obtain future deformations, instead of synthesizing images directly. The method combines a VoxelMorph style U-Net with FiLM based metadata conditioning and is evaluated on the OASIS-3 dataset, with comparisons to VoxelMorph and a stationary SVF baseline.
Experimental results suggest that metadata conditioned deformation prediction captures nonlinear, clinically meaningful aging and disease-related trends, particularly in ventricular and hippocampal regions. The importan part of the paper lies in shifting the focus from image-level synthesis to interpretable geometric forecasting.

**Strengths:**

- Framing the task as prediction of deformation fields rather than images aligns well with how longitudinal change is typically quantified in neuroimaging studies.

- The methodological choices sound relevant, the use of diffeomorphic SVFs and scaling-and-squaring, which ensures topological consistency and interpretability of the predicted transformations.

- Conditioning the deformation model on metadata via FiLM is technically well, + the comparison between the fully conditioned model and the time-scaled stationary baseline is well thought out.

- The analysis goes beyond image similarity metrics and includes deformation trends, hippocampal volume changes, and subgroup analyses (CN vs AD), which really strengthens the biological plausibility of the results of the paper

**Weaknesses:**

The training setup depends on paired longitudinal scans and a registration-style image similarity objective, which means that the model is learning future deformations indirectly through observed follow-up images.  Tthe paper does not clearly characterize how robust the learned deformation model is to shifts in the distribution of time intervals, age ranges, or disease severity, nor does it quantify performance degradation when extrapolating beyond the dominant regimes present in OASIS-3 dataset

The evaluation is richer than in standard image-synthesis work, but still relies heavily on agreement with VoxelMorph as a reference. Since VoxelMorph solves a different, better-posed problem (registration with access to both images), consistency with its outputs does not necessarily imply that the predicted future deformations are correct or unique. Possible more deformation-specific or anatomy-driven validation criteria would strengthen the argument that the predicted fields are biologically plausible rather than smooth, metadata-conditioned averages.

**Detailed Comments:**

An ablation study examining the contribution of individual metadata variables (e.g., removing diagnosis, age, or sex one at a time) would strengthen the claim that the conditioning mechanism captures meaningful subject-level effects beyond time interval alone.

Overall, these weaknesses and comments are largely about scope, validation depth, and clarity rather than fundamental flaws. Addressing them would further strengthen an already solid and well-executed contribution.

**Justification Of Final Rating:**

The authors have addressed my comments, the added analyses on sensitivity to training data distributions and the metadata/anatomy shuffling experiments significantly strengthen the paper’s claims about robustness and subject-specificity, and the new ablation results in the appendix provide helpful clarity on the role of individual conditioning variables. Minor text edits were done by authors to improve readability and other reviwers comments were also integrated intto the manuscript. Overall it sounds like a strong paper with clear contribution.

**Justification Of The Preliminary Rating:**

This paper makes a well-motivated contribution that advances how longitudinal imaging prediction can be formulated and studied. The idea of forecasting future diffeomorphic deformation fields from a single baseline image and metadata is both novel in perspective and well aligned with how anatomical change is interpreted in clinical and research settings. While the individual components of the framework such as SVFb ased diffeomorphic modeling, FiLM conditioning, and VoxelMorph-style training are known, their integration into a coherent framework for “registration with a missing fixed image” is thoughtfully executed.

Rather than relying only on image synthesis metrics, the authors examine deformation trends, age and interval dependencies, hippocampal volume trajectories, and subgroup effects, demonstrating a strong connection between the model’s outputs and established neurobiological expectations. The comparison with a stationary SVF baseline shows why simple timebscaled integration is insufficient for realistic longitudinal forecasting.

Importantly, the paper is honest about its limitations, especially regarding stationarity assumptions, data imbalance, and uncertainty, and supports these discussions with both theoretical analysis and empirical evidence. The writing is clear, the structure is logical, and the methodological choices are well justified, the figures nicely support the discussion.

**Questions To Address In The Rebuttal:**

How sensitive is FutureMorph to the distribution of time intervals and ages seen during training, and how does it perform when extrapolating to longer intervals or age ranges not well represented in the OASIS?

Is there any evidence that the predicted deformation fields are not simply learned averages conditioned on time and diagnosis, but genuinely subject-specific beyond what is encoded in the baseline image?

Are there deformation-specific validation strategies, beyond agreement with VoxelMorph, that could strengthen confidence in the biological plausibility of the forecasts?

---

> ### Author Response · Authors · 2026-01-23
> **Response to Reviewer R7UD (1/2)**
>
> We thank the reviewer for the positive feedback and kind remarks, and we have addressed the points raised and the suggested directions. (All modifications are highlighted in red in the revised manuscript.)
>
> **Weaknesses**:
>
> **W1**: The paper does not clearly characterize how robust the learned deformation model is to shifts in the distribution of time intervals, age ranges, or disease severity, nor does it quantify performance degradation when extrapolating beyond the dominant regimes present in the OASIS-3 dataset.
>
> **A1**: We thank the reviewer for raising this important point. True out-of-distribution robustness beyond the regimes present in OASIS-3 cannot be directly assessed due to the lack of ground-truth follow-up scans in those settings. However, we analyze sensitivity to distributional imbalance within the training data by stratifying test cases into age and time-interval bins and relating Dice scores to the corresponding training-set counts. We observe that performance degrades most noticeably for time-interval ranges that are underrepresented during training, whereas age has a weaker effect. We have added this analysis and discussion in the new *Sensitivity to Training Data Distributions paragraph* in the Results and Discussion section and summarized results of this study in **Figure 8**.
>
> **W2**: The evaluation is richer than in standard image-synthesis work, but still relies heavily on agreement with VoxelMorph as a reference. Since VoxelMorph solves a different, better-posed problem (registration with access to both images), consistency with its outputs does not necessarily imply that the predicted future deformations are correct or unique. Possible more deformation-specific or anatomy-driven validation criteria would strengthen the argument that the predicted fields are biologically plausible rather than smooth, metadata-conditioned averages.
>
> **A2**: We thank the reviewer for the positive feedback and interesting comments. Because ground-truth deformation fields are unavailable, we use VoxelMorph as reference standard only as a sanity check and assess agreement via hippocampal Jaccobian determinants and regional deformation statistics. In the revised manuscript, we evaluate sensitivity to metadata and anatomy shuffling to verify that FutureMorph produces subject-specific rather than population-average predictions; the results are reported in **Table 4 (Appendix D)**. These additions are discussed in the revised Results and Discussion sections under Sensitivity to Shuffling and Consistency with VoxelMorph.
>
> **Detailed Comments:**
>
> **DC**: An ablation study examining the contribution of individual metadata variables (e.g., removing diagnosis, age, or sex one at a time) would strengthen the claim that the conditioning mechanism captures meaningful subject-level effects beyond time interval alone.
>
> **A**: We thank the reviewer for this suggestion. We performed an ablation study by shuffling individual metadata variables (diagnosis, age, or time interval) separately. As reported in **Table 3 (Appendix D)**, Dice scores drop more for age and time interval than for diagnosis, confirming that these factors are important for conditioning and that FutureMorph captures meaningful subject-specific effects beyond time interval alone. We elaborate on this in the sensitivity to shuffling, Results and Discussion section: *”We performed an ablation study to evaluate the contribution of individual metadata variables by shuffling diagnosis, age, or time interval separately. **Table 3** in **Appendix D**  reports Dice scores for each case, showing larger drops for age and time interval than for diagnosis. These results indicate that these factors are particularly important for conditioning and support the claim that FutureMorph captures meaningful subject-specific effects beyond temporal spacing alone.*”

---

> > ### Comment · Reviewer_R7UD · 2026-01-31
> >
> > Thank you for adding sensitivity to data distributaion and shuffling parts, it definitely strenfthens your claims. Overall I am quite happy with Appendix D with its ablation. Explanation about VoxelMorph usage is reasonable.

---

> ### Author Response · Authors · 2026-01-23
> **Response to Reviewer R7UD (2/2)**
>
> **Question to address**
>
> **Q1**: How sensitive is FutureMorph to the distribution of time intervals and ages seen during training, and how does it perform when extrapolating to longer intervals or age ranges not well represented in the OASIS?
>
> *A1*: We thank the reviewer for this suggestion. While true out-of-distribution performance cannot be assessed, we analyzed sensitivity within the training data and found that Dice scores drop most for underrepresented time-interval ranges, with age having a weaker effect. We added **Figure 8** to illustrate the dice and counts matrices and elaborated on these outcomes in the Sensitivity to Training Data Distributions paragraph: *“We examined how sensitive FutureMorph is to the distributions of age and time intervals observed during training. Test cases were grouped into age bins (2-year intervals and decade ranges) and time-interval bins, and Dice scores were computed per group alongside the corresponding training-set counts. As shown in Figure 8, the performance degraded noticeably for time-interval ranges that were underrepresented in the training data, whereas age distributions had a weaker effect. This indicates that the time interval between visits is a dominant factor for prediction accuracy, consistent with our modeling assumptions.”*
>
> **Q2**: Is there any evidence that the predicted deformation fields are not simply learned averages conditioned on time and diagnosis, but genuinely subject-specific beyond what is encoded in the baseline image?
>
> **A2**: We thank the reviewer for this question. To verify that FutureMorph predictions are subject-specific, we performed controlled shuffling of metadata and baseline anatomy. Dice scores dropped substantially when metadata was shuffled, and intensity-based metrics degraded when anatomy was shuffled, confirming that FutureMorph leverages both subject-specific anatomy and metadata conditioning rather than producing population-average predictions. We added **Table 4 in Appendix D** of the revised paper; we mentioned the detailed discussion in  **“Sensitivity to Shuffling”** paragraph in Results and Discussions: *“For metadata shuffling, we permuted the conditioning variables while keeping the anatomy fixed. For anatomy shuffling, we replaced the moving image with that of another subject matched for age and disease class while keeping the segmentation unchanged. Metadata permutation substantially degraded Dice (See Table 4, Appendix D), whereas anatomy shuffling primarily affected intensity-based metrics. The anatomy shuffle produced the largest overall degradation with milder effects on Dice, indicating that metadata mainly governs anatomical consistency while baseline anatomy drives intensity fidelity. These results indicate that FutureMorph leverages both subject-specific anatomy and metadata conditioning rather than population-average priors.”*
>
> **Q3**: Are there deformation-specific validation strategies, beyond agreement with VoxelMorph, that could strengthen confidence in the biological plausibility of the forecasts?
>
> **A3**: We thank the reviewer for this suggestion. While ground-truth deformations are unavailable, beyond agreement with VoxelMorph we validate our forecasts using predicted hippocampal change rates, which aligns with VoxelMorph and known neurodegenerative trends. Future work could strengthen validation by extending analyses to multiple brain regions or by evaluating the predicted future images using a classifier—trained or pre-trained—to assess whether clinical labels (e.g., CN vs AD) are preserved, providing additional evidence of biologically plausible forecasts. In the modified paper, we added these in the **conclusion**: *“Since there is no direct way to validate our predictions, we used VoxelMorph as a reference to assess agreement of FutureMorph forecasts with registration-based deformations. Future work could incorporate classifier-based guidance on predicted future images or volume-based loss to further support the biological plausibility of the forecasts.”*

---

### Author Rebuttal · Authors · 2026-01-23

**Rebuttal:**

We thank all reviewers for their insightful feedback and have addressed their comments to the best of our ability, extending the ablations and experiments as suggested. For a full response, please see the individual replies to each reviewer. All corresponding changes are highlighted in red in the revised manuscript.

**Supporting Material:**

/attachment/9d7451228f7ca4f01b1e4bda32174e2df9390964.pdf

---

### Comment · Area_Chair_MmhZ · 2026-01-26
**Start of Discussion Phase**

Thanks to the authors for providing a detailed rebuttal and for engaging constructively with the reviewers’ comments.

I kindly ask all reviewers to carefully read the rebuttal and assess whether the authors’ responses sufficiently address the raised concerns.
If aspects remain unclear or require further clarification, please use the official discussion/comments function to ask follow-up questions and engage in discussion with the authors during this phase. Reviewers are also welcome to comment on and respond to the reviews and rebuttal points raised by other reviewers, where relevant.

Please note that by the end of the discussion period, all reviewers are expected to verify and, if appropriate, update their scores. Even if you agree with the authors’ responses but decide not to change your rating, please leave a brief comment indicating that you have read and considered the rebuttal.

---

### Meta-Review · Area_Chair_MmhZ · 2026-02-09

**Recommendation:** Accept (Oral)
**Confidence:** 4

**Metareview:**

Justification:
Two reviewers recommended strong accept and one weak accept, and all reviewers agree that the work merits oral presentation. The paper might be considered for the special issue.
The paper presents a technically sound and conceptually interesting approach; while it is evaluated on a relatively small, single dataset and does not focus on clinical application or usability, it introduces a novel perspective on longitudinal modeling that is well suited for presentation and discussion at the MIDL conference.

A summary of the strength and remaining weaknesses below:

**Summary:**

The paper introduces FutureMorph, a method for forecasting future diffeomorphic deformations in longitudinal MRI from a baseline scan and subject metadata. The approach frames longitudinal prediction as registration with a missing fixed image, using a metadata-conditioned U-Net to predict stationary velocity fields.

**Strengths:**
1.	Interesting topic with a clear and well-motivated research question, supported by a solid and appropriate literature review.
2.	Technically sound methodology, with it seems effective metadata conditioning (via FiLM) and a well-designed comparison to a time-scaled stationary baseline.
3.	Reformulates longitudinal prediction as forecasting diffeomorphic deformation fields rather than image synthesis.
4.	Evaluation beyond image similarity metrics, including deformation trends, hippocampal volume changes, and subgroup analyses (e.g., CN vs AD), supporting biological plausibility.

**Weaknesses that remain after the rebuttal:**

1.	Experimental validation is restricted to a single dataset (OASIS-3), limiting evidence for generalizability.
2.	The training and evaluation setup only includes subjects with exactly two visits, excluding patients with longer longitudinal trajectories and thereby limiting assessment of multi-timepoint consistency.
3.	The relevance and interpretability of deformation magnitude as a primary evaluation metric remain unclear, particularly in relation to established longitudinal morphometric measures.
4.	The Alzheimer’s disease cohort is relatively small, which limits statistical power and leads to weak or noisy correlations (e.g., deformation magnitude versus age).

---

### Decision · Program_Chairs · 2026-02-13

Accept (Poster)